# Comparative Analysis of CT and MRI Combined with RNA Sequencing for Radiogenomic Staging of Bladder Cancer

**DOI:** 10.3390/ijms26199570

**Published:** 2025-09-30

**Authors:** Joshua Levy, Toru Sakatani, Kaoru Murakami, Yuki Kita, Takashi Kobayashi, Susan Win, Saro Manoukian, Charles J. Rosser, Hideki Furuya

**Affiliations:** 1Samuel Oschin Comprehensive Cancer Institute, Cedars-Sinai Medical Center, Los Angeles, CA 90048, USA; joshua.levy@cshs.org (J.L.); charles.rosser@cshs.org (C.J.R.); 2Pathology and Laboratory Medicine, Cedars-Sinai Medical Center, Los Angeles, CA 90048, USA; 3Department of Computational Biomedicine, Cedars-Sinai Medical Center, Los Angeles, CA 90069, USA; 4Department of Urology, Graduate School of Medicine, Kyoto University, Kyoto 606-8507, Japan; sktntoru5@kuhp.kyoto-u.ac.jp (T.S.); kaorum@kuhp.kyoto-u.ac.jp (K.M.); kitayuki@kuhp.kyoto-u.ac.jp (Y.K.); selecao@kuhp.kyoto-u.ac.jp (T.K.); 5Department of Radiology, Cedars-Sinai Medical Center, Los Angeles, CA 90048, USAsaro.manoukian@cshs.org (S.M.); 6Department of Urology, Cedars-Sinai Medical Center, Los Angeles, CA 90048, USA; 7Department of Biomedical Science, Cedars-Sinai Medical Center, Los Angeles, CA 90048, USA

**Keywords:** muscle-invasive bladder cancer, radiogenomics, artificial intelligence, CT, MRI, RNASeq

## Abstract

Accurate staging of bladder cancer (BCa) is important for identifying optimal treatment. Currently, clinical tumor staging for BCa relies on computed tomography (CT) scans, but these can lead to under- or overstaging of patients. Recent research suggests that using magnetic resonance imaging (MRI) along with RNA sequencing (RNASeq) gene expression analysis can provide more precise staging. In this study, 31 matched CT scans, MRI images, and formalin-fixed, paraffin-embedded (FFPE) tissues were collected. First, two radiologists reviewed the images for staging BCa. Next, radiomics features were extracted from both CT and MR images, and computational radiogenomics analyses were performed. Subsequently, RNASeq was performed using FFPE tissues of TURBT prior to cystectomy. A radiogenomic analysis was conducted to identify advanced T-stage signatures. Regarding imaging alone, MRI was found to be more accurate in staging >T2 compared to CT scans. Within a retrospective cohort, MRI radiogenomic signatures were more effective in staging patients than CT, with genomic features playing a significant role. Using canonical correlation analysis, we additionally identified radiomic features underlying genomic signatures of advanced tumor stage. When applying these signatures across a small prospective cohort, MRI radiomic data were able to stratify stage; however, the addition of the same genomic features did not improve the sensitivity and specificity of the model. These preliminary results are promising, but additional research with larger sample sizes is needed to draw definitive conclusions and explore further correlations and statistical interactions between genes and imaging features through machine learning techniques as we move radiogenomics to the clinic.

## 1. Introduction

Muscle-invasive bladder cancer (MIBC) comprises approximately 25% of bladder cancer (BCa) and is associated with significant morbidity and mortality [1]. In the US, radical cystectomy remains the treatment of choice for MIBC, while we have noticed increased rates of bladder preservation therapy [2]. Indeed, a patient’s quality of life could be negatively affected by radical cystectomy [3]; thus, there is continued intrigue in bladder preservation therapy. Multiple bladder preservation options exist, although the approach of maximal transurethral resection of bladder tumor (TURBT) performed along with concomitant chemoradiation therapy is the most favored [4,5]. A critical factor in planning either radical cystectomy or bladder preservation therapy is accurate staging of the tumor. If the BCa is not MIBC, then the patient may be treated with local (i.e., intra-vesical) agents. The current NCCN guidelines recommend TURBT to obtain tissue to diagnose bladder cancer and contrast-enhanced computed tomography (CT) of the abdomen and pelvis at the time of bladder cancer diagnosis to assess the local extent of the cancer, including regional lymph nodes. However, 1/3 of patients undergoing these CT scans are understaged, while another 1/3 of patients are overstaged according to the final pathological stage [6]. Thus, a more accurate non-invasive staging modality is necessary, which could improve medical decision-making between various treatments (e.g., radical cystectomy vs. bladder preservation).

Radiogenomics is an emerging area of research that has developed due to the capacity to link radiologic imaging information (radiomics) with genomic data from tumors, thereby enhancing the understanding of tumor phenotypes and improving the prediction of clinical outcomes [7]. Radiomics is a relatively new area deploying a quantitative approach to medical imaging by extracting a large number of features from medical images, such as CT scans and magnetic resonance imaging (MRI), using data characterization algorithms [8,9]. This could not only allow us to diagnose cancer but also predict clinical outcomes and treatment responses. On the other hand, genomic medicine is an established medical discipline that involves using genomic information from a patient as part of their clinical care (e.g., for diagnostic or therapeutic decision-making) [10]. Since the Radiogenomics Consortium was established in the United Kingdom in 2009 [11], the results from radiogenomic studies have reported a strong association between radiomic parameters and genomic features, e.g., DNA mutations, mRNA expression, and copy number variations [12]. Imaging parameters with strong associations to genomic data may serve as reliable markers for tumor diagnosis and evaluation (i.e., staging), prognosis, and treatment response. In addition, integrating radiogenomic data may (a) allow for the exploration of molecular interconnections that explain various tumor features captured by the radiomic parameters and (b) validate the mechanistic pathways supporting such interconnections, thereby allowing us to gain a deeper understanding of the cancer and its pathophysiology. Taken together, we hypothesize that radiogenomics can help tailor treatments to individual patients with BCa, identifying those who may benefit most from specific therapies (e.g., radical cystectomy vs. bladder preservation) as described above.

Currently, when conducting a PubMed literature search for radiogenomics and BCa, only four articles, including ours, are available. Outside of BCa, feasibility studies in small but well-characterized cohorts have demonstrated the potential of radiogenomic integration, including differentiation of renal tumor subtypes [13] and multimodal radiopathomics for therapy response prediction in NSCLC [14]. In the first, Lin et al. reported the results of RNA sequencing (RNASeq) data, radiomics features, and clinical parameters of 62 BCa patients and found that the radiomics and transcriptomics signatures significantly stratified patients into high- and low-risk groups in terms of progression-free survival (PFS) [15]. A second study discovered N6-Methyladenosine subtypes that predict clinical outcomes and are linked to immune response markers like PD1 and CTLA4 [16]. Xu et al. also found correlations between angiogenesis gene mutations and BCa patient characteristics, outcomes, and immune cell presence [17]. Furthermore, radiogenomics in our study, comprising cohorts, showed promise in evaluating BCa staging, achieving 94% sensitivity, 88% specificity, and 92% accuracy in distinguishing intra- from extra-BCa stages [18].

Multiparametric MRI (mpMRI) is emerging as the imaging modality of choice in tumor staging, with a reported sensitivity of 90% in detecting BCa and 76% sensitivity with 89% specificity in detecting metastatic lymph nodes [19]. The superior soft tissue contrast and versatile imaging sequences of MRI can facilitate margin definitions critical in cystectomy and bladder preservation strategies. Leading the way in transitioning from CT scans to MRI with regard to cancer stage is prostate cancer. Accumulating evidence in prostate cancer suggested that the image quality of 3 T MRI was superior to 1.5 T MR [20,21], and both are superior to CT imaging. In fact, MR imaging of the prostate has replaced CT scan imaging in US, European, and Asian prostate cancer management guidelines [22,23,24]. Similarly, studies in BCa also demonstrated that the use of 3 T MRI achieves higher specificity than 1.5 T MRI in the local evaluation of BCa [25]. With these images, extensive quantitative imaging metrics can be collected. Thus, we believe that 3 T mpMRI offers an opportunity to reduce staging errors through better anatomical visualization [25,26] compared to traditional CT imaging [27,28]. In parallel, deep learning approaches have recently been explored to enhance MRI-based staging performance in BCa, demonstrating promising early results [29,30]. Moreover, advanced imaging modalities, such as FDG-PET/CT, have shown utility in specific contexts, including detecting lymph node involvement in patients with variant histology of MIBC [31]. In this study, we compared the performance of CT scan and MRI in staging BCa by traditional radiologist review and radiogenomic in combination with RNASeq data from formalin-fixed, paraffin-embedded (FFPE) tissues using a retrospective cohort. Then, with MRI/RNASeq-based radiogenomics appearing superior, we evaluated its performance in a pilot prospective study in BCa.

## 2. Results

### 2.1. Comparison of Radiologic Interpretation on CT and MR Images

The retrospective cohort (Kyoto University, *n* = 31) included patients across Ta–T4 stages, with a predominance of muscle-invasive cases (T2–T4, 74%). The prospective cohort (Cedars-Sinai, *n* = 10) included both non-muscle-invasive BCa (NMIBC, Ta–T1, 50%) and MIBC (T2–T4, 50%) patients, as detailed in Materials and Methods. Figure 1 illustrates a CT and MR image of the bladder from the same subject. As is clear, MRI resolution is superior to CT, allowing the visualization of a tumor in the right posterolateral bladder wall. Next, all CT and MR images were reviewed by at least two radiologists to generate a consensus to stage BCa. As shown in Table 1, the accuracy of staging ≤pT1 and pT2 was the same between CT and MR images (60% and 50%, respectively). However, interestingly, the accuracy of staging ≥pT3 is better in MRI than in a CT image (57% vs. 35%).

### 2.2. Transcriptional and Radiomic Correlates of Advanced Stage

In our retrospective cohort, after accounting for age and sex, genes were identified with expression that increased or decreased with advanced stage. Genes upregulated in extra-vesical BCa corresponded to pathways characterized by fibroblast growth factor receptor (FGFR) signaling and RNA metabolism, whereas genes related to epidermal growth factor receptor (EGFR)/rat sarcoma virus (RAS)/mitogen-activated protein kinase 1 (MAPK1) signaling were associated with intra-vesical BCa (Appendix A; Figure 2).

We also identified a number of radiomic features (CT/MR) associated with tumor staging (see Appendix A). Using a bin count of 16, MRI features negatively associated with advanced stage included GLCM autocorrelation and high gray-level emphasis, which reflects loss of structured high-intensity features. MRI features correlated with increasing stage included low gray-level emphasis features, corroborating the prior findings. For the same bin count size, *ShortRunHighGrayLevelEmphasis* CT features (abundance of smaller high-intensity patterns) were negatively correlated with advanced stage, suggesting loss of high-intensity concentrated patterns. Intra-vesical tumors were more spherical (*Shape*_*Sphericity*) and defined by greater contrast or texture variation (*GrayLevelVariance*, *ClusterShade*). Extra-vesical tumors were more elongated (*MajorAxisLength*) and contained smaller low-intensity regions in the tumor (*SmallAreaLowGrayLevelEmphasis*).

### 2.3. Canonical Correlation Uncovers Biological Pathways Underlying Progression-Related Radiomic Features

We identified canonical correlation components associated with advanced tumor stage (CT: canonical correlation analysis [CCA]1 [ρ = 0.13, *p* = 0.372], CCA3 [ρ = −0.37, *p* = 0.0182], MRI: CCA4 [ρ = 0.29, *p* = 0.0099], CCA1 [ρ = −0.38, *p* = 0.0025]), comprising gene and radiomic features (Figure 3; Appendix A). While CT-CCA1 is not significantly associated with stage, its radiomic and genomic projections remain well aligned (canonical correlation r = 0.633, *p* = 3.93 × 10^−5^), as do those of CT-CCA3 (r = 0.680, *p* = 9.70 × 10^−5^), MRI-CCA1 (r = 0.684, *p* = 8.32 × 10^−5^), and MRI-CCA4 (r = 0.655, *p* = 2.07 × 10^−4^), indicating strong cross-modal associations across components. The genes and pathways linked to these components overlapped with those identified in our transcriptome-wide association study, while the radiomic features within these CCA components mirrored those that stratified stage in univariable analysis. For instance, this included associations between low gray-level emphasis features, FGFR-related pathways, advanced stage (MRI-CCA1), and variance in intensities and EGFR signaling associated with lower stage (MRI-CCA4).

Figure 4 illustrates a CT and MR image of the bladder from the same subject. As is clearly evident, MRI resolution is superior to CT, allowing for the visualization of a tumor in the right posterolateral bladder wall, which is consistent with radiologic interpretation (Figure 1). The AIC and BIC calculations reveal that MRI consistently demonstrates lower values compared to CT, indicating an improved model fit (Figure 4). While RNA features alone provided a suboptimal fit, the integration of RNASeq data further enhanced the performance. Likelihood ratio testing confirmed the performance improvements achieved by incorporating MRI features (Appendix A). Across various bin numbers, the addition of MRI-derived features to CT significantly improved the likelihood (*p* < 0.05), highlighting the complementary value of MRI. Conversely, adding CT features to MRI did not result in significant improvements, or at least the effect was less pronounced, regardless of whether RNASeq features were included. When modeling with gene expression data, the enhancement in performance remained consistent with the trends observed, reinforcing the superior contribution of MRI features over CT (Appendix A). Notably, for MRI-derived features, increasing the number of voxel bins for radiogenomics feature calculation—leveraging the higher resolution of MRI—further improved the fit. Furthermore, cross-validation within the retrospective cohort using radiogenomic PCA features demonstrated that the MRI–gene model outperformed the CT–gene model in predicting tumor stage, with the MRI–gene model achieving an AUC of 0.833 ± 0.212 compared to 0.758 ± 0.193 for the CT–gene model. Thus, we selected MRI features calculated from 16 bins as the final feature set for model development and integration with RNASeq.

### 2.4. A Pilot Study Developing Workflow for Radiogenomic Models

Multivariable regression models were fit and cross-validated on the retrospective cohort and then applied to the prospective cohort to predict advancing stage from radiomic and radiogenomic features. These models demonstrated some overlap in radiomic features found to be important for prediction (Figure 5A). In our prospective cohort, MRI features alone, identified in the retrospective cohort, were highly predictive of advanced stage (Figure 5B, Table 2). Specifically, the MRI alone model achieved an AUC of 0.84 ± 0.22 (SE), with a sensitivity of 0.84 ± 0.17 and a specificity of 0.89 ± 0.16. By comparison, the MRI + RNASeq model achieved an AUC of 0.75 ± 0.22, with a sensitivity of 0.80 ± 0.19 and a specificity of 0.78 ± 0.2. Notably, cross-validation performance within the retrospective cohort (MRI alone AUC = 0.77 ± 0.14) closely matched that observed in the prospective cohort (Table 2), supporting the selected features and model.

## 3. Discussion

This exploratory study aimed to characterize radiogenomic features indicative of advanced tumor stage in BCa, leveraging canonical correlation analysis to identify gene signatures and biological processes that correlate with radiologic features, which in turn correlate with tumor progression. Our retrospective cohort analysis revealed that MRI features were significantly more predictive of advanced tumor stage than CT features, with their predictive power further enhanced through integration with gene expression data, while our prospective pilot study demonstrated that radiomic features associated with MRI alone were highly predictive of advanced stage.

It is important to note that the retrospective cohort (Kyoto University, *n* = 31) consisted exclusively of Asian patients with a predominance of higher-stage tumors (T2–T4), whereas the prospective cohort (Cedars-Sinai, *n* = 10) included a more racially diverse population (seven Caucasian, two Hispanic, and one Asian) and was enriched for earlier-stage tumors (Ta–T2). These demographic and stage-related imbalances limit the generalizability of our findings and may confound the assessment of the added value of RNASeq data. Given the small sample sizes, subgroup sensitivity analyses were underpowfered and not feasible; however, MRI radiomic features demonstrated consistent performance across both cohorts, suggesting their potential robustness. However, larger multi-center studies with greater diversity and balanced stage distribution will be required to validate radiogenomic signatures. Indeed, as highlighted in prior systematic reviews of radiomics in BCa, the vast majority of published studies lack independent external validation, limiting generalizability [32]. This gap is even more pronounced for radiogenomic approaches, where virtually no studies have yet incorporated multi-institutional validation. An important limitation of our study is the small sample size of both retrospective (*n* = 31) and prospective (*n* = 10) cohorts. Small cohorts inherently increase the risk of both false-positive and false-negative associations, reduce the stability of radiomic feature selection, and limit the generalizability of multivariable models. In addition, subgroup analyses to evaluate the effects of demographic or tumor-related variables could not be performed due to limited power. Based on effect sizes observed in this pilot analysis, we estimate that a cohort of several hundred patients across multiple institutions would be required to achieve approximately 80% statistical power (α = 0.05) to validate the observed differences in staging accuracy between radiogenomic and imaging-only models. Such large-scale, multi-center studies will be critical to establish robust and generalizable radiogenomic signatures for BCa staging.

Through both univariable and multivariable analyses, MRI and CT radiomic features were identified as significantly associated with tumor stage. Advanced-stage tumors demonstrated radiomic signatures characterized by lower sphericity, greater elongation, increased low gray-level emphasis and autocorrelation, and reduced high gray-level emphasis. Additionally, these tumors exhibited decreased intensity skewness, entropy, and complexity—textural patterns commonly linked to invasive and infiltrative growth. These findings are consistent with multiple prior radiomics studies that have reported similar shape [33] and texture features (e.g., increased *LowGrayEmphasis*) to be associated with MIBC [32,34] compared to NMIBC [35], as well as with higher histologic grade [36]. A number of studies have developed radiomic signatures associated with BCa progression. While an exhaustive comparison between our identified features and those reported in prior studies—particularly from our univariable analysis—is beyond the scope of this work, we encourage readers to consult the systematic reviews cited herein [32,34]. Notably, many of these studies do not disclose their exact feature sets; however, where information was available, we observed general concordance with our findings.

CCA further identified radiomic features that not only correlate with tumor stage but also reflect distinct molecular pathways involved in tumor progression [32,37]. These morphologic changes may be linked to fibroblast-driven tumor spread [37], as CCA revealed that both FGFR pathway activation and radiologic features, such as *LowGrayLevelRunEmphasis,* were included in a shared CCA component associated with tumor progression. This aligns with the role of FGFR signaling in promoting tumor–stromal interactions, epithelial–mesenchymal transition (EMT), and invasion, which may contribute to fibrotic remodeling. Additionally, this fibrotic microenvironment could explain the observed change in small, low gray-level areas, difference in structured high-gray features [34,35], and altered kurtosis—potentially reflective of necrosis, hypoxia, and stromal interactions.

Importantly, these radiogenomic associations are consistent with known biological mechanisms in BCa. FGFR3 mutations and pathway activation are well recognized in both non-muscle-invasive and invasive disease, where they promote stromal remodeling and epithelial–mesenchymal transition, contributing to progression and invasion. Conversely, EGFR/RAS/MAPK signaling is often activated in early-stage tumors and is associated with epithelial proliferation, aligning with radiomic features of organized and homogeneous tumor morphology. These findings are also concordant with established molecular subtype classifications of BCa, in which luminal–papillary tumors frequently harbor FGFR3 alterations [37,38,39], while basal/squamous subtypes are enriched for EGFR pathway activation [38,39]. Taken together, these results suggest that radiomic signatures may serve as non-invasive proxies for canonical pathways underlying BCa progression.

Conversely, a separate CCA component linked EGFR/RAS/MAPK pathway activation to radiologic features characteristic of earlier-stage tumors, which appeared more homogeneous in shape. These features, such as high gray-level textures (e.g., *HighGrayLevelZoneEmphasis*) and structured intensity variation (e.g., *Complexity*), are indicative of increased cellularity (e.g., *RobustMeanAbsoluteDeviation*), reflecting more organized tumor architecture. This suggests that intra-vesical tumors may be driven more by an early EGFR-driven proliferation through initiating alterations than by FGFR-associated stromal remodeling and invasion.

Importantly, these findings suggest that radiologic signatures can serve as proxies for key molecular pathways involved in tumor progression, providing a non-invasive means of assessing tumor biology. While this study primarily identified radiogenomic associations linked to tumor stage, future work will focus on deriving more complex relationships beyond T-stage alone, integrating additional biological and clinical factors. Further validation through spatial mapping of key gene and protein signatures with MRI/CT features will be critical in determining whether these localized texture patterns directly reflect molecular pathways, thereby establishing their potential as robust non-invasive biomarkers.

The performance of the radiomics model in our prospective cohort (AUC = 0.84), despite sample size limitations, was consistent with previous reports in the literature [18]. However, we were surprised to find that the RNA features did not offer additional predictive value to our 4D mpMRI MR. This may be attributed to several limitations, most notably, limited sample size, demographic imbalances, and the possibility of a significant batch effect, all of which could lead to overfitting or multicollinearity. First, the retrospective cohort predominantly consists of Asian patients, while the prospective cohort is largely composed of Caucasians. Second, the majority of patients in the retrospective cohort were diagnosed with stages T2–4, whereas the prospective cohort primarily included patients with stages Ta, T1, and T2. As a result, in the retrospective cohort, we compared intra-vesical (Ta, Tis, T1, and T2) versus extra-vesical (T3 and T4) tumors, while in the prospective cohort, we compared NMIBC (Ta, Tis, and T1) versus MIBC (T2). RNA extracted from FFPE specimens was of variable quality in these cohorts, which may have reduced the stability of transcriptomic associations. All RNA-seq assays, however, were performed simultaneously using the same workflow, thereby minimizing the risk of technical batch effects. Thus, given these significant limitations, it is crucial to emphasize the exploratory nature of this study and caution against over-interpreting the results or biomarkers identified from the multivariable model. Importantly, the lack of incremental predictive value from RNA features in the prospective cohort should be interpreted not only as a limitation but also as a critical result of this study. MRI-derived radiomic features demonstrated consistent performance across both cohorts, suggesting that they may represent a more robust and reproducible modality for staging. In contrast, transcriptomic features derived from FFPE tissue were likely affected by sample quality and cohort heterogeneity, reducing their reliability. This finding highlights that while radiogenomic integration holds promise, the incremental value of RNASeq may only become evident in larger multi-institutional studies, ideally using fresh-frozen material to minimize technical variability. As such, our results provide guidance for future trial design by indicating that MRI radiomics should be prioritized as a robust foundation upon which transcriptomic data can be layered once validated in larger datasets. Our feasibility-stage approach is consistent with prior radiogenomics and multimodal studies conducted in small cohorts, such as those exploring renal tumor differentiation (*n* = 14) [13,14]. Sequencing QC metrics also demonstrated variability (Appendix A), underscoring the need for larger cohorts and fresh-frozen tissues in future studies. Future studies with larger cohorts and fresh-frozen tissues will be important to minimize technical artifacts and confirm the stability of radiogenomic associations. Instead, we recommend focusing on the comparative assessment of AIC/BIC between MRI and CT features as an early finding that could motivate additional study of MRI features for radiogenomic analyses in the context of BCa prognostication.

However, it is encouraging that the radiomic analysis of MRI achieved a relatively high AUC in the prospective cohort, suggesting that the data analysis approach is valid and supports further investigation. We would expect radiologic imaging features to be less encumbered by batch effect, hence the more consistent performance across the retrospective and prospective cohorts. For future studies, a multi-center approach could be beneficial to enhance sample size and diversity. Increasing our study cohort can permit the introduction of modeling approaches that can reduce batch effect and improve generalization of RNA markers and the adoption of machine learning strategies for improved predictive accuracy. Several practical considerations must be addressed before radiogenomics can be integrated into clinical workflows. These include sequencing costs, longer turnaround times compared with imaging alone, and the need for adequate tissue from TURBT specimens. In addition, standardized analytical pipelines and multi-institutional validation will be necessary to establish reproducibility and generalizability. While the present study was not designed to resolve these issues, it highlights the potential of MRI radiomic features and sets the stage for future work to define how transcriptomic data may best be incorporated.

## 4. Materials and Methods

### 4.1. Retrospective Study Cohort

The study was performed after approval by the Cedars-Sinai Medical Center Institutional Review Board (IRB) (STUDY00001310) and Kyoto University IRB (#G1301) under a request for waiver of consent on archived pathologic specimens and imaging data (CT and MRI). A retrospective cohort comprising 31 matched CT scans, MRI, and formalin-fixed paraffin-embedded (FFPE) tissues from TURBT were collected from Kyoto University and used as a cohort (Table 3). The study subjects were identified for the previous study as described previously [18]. Patients selected for this cohort were those who underwent radical cystectomy without neoadjuvant chemotherapy. The final cystectomy pathology served as the reference standard for tumor stage. RNASeq data was obtained from the study. Due to low RNA concentration or low % DV200 (Appendix A), two subjects were excluded from radiogenomics analysis. RNA quality and sequencing QC metrics, including concentration and DV200 (percentage of RNA fragments > 200 nucleotides), total reads, and mapping percentages, are reported in Appendix A. Sequencing QC metrics, including total reads and mapping percentages, are also provided. As expected with FFPE-derived RNA, quality was variable across samples. This cohort, therefore, included a spectrum of stages (Ta–T4), with the majority (74%) being muscle-invasive (T2–T4).

### 4.2. Prospective Study Cohort

The study was conducted after approval by the Cedars-Sinai Medical Center IRB (STUDY00001296) according to the Declaration of Helsinki and Good Clinical Practice guidelines. Inclusion criteria required radical cystectomy without neoadjuvant chemotherapy, and the final cystectomy pathology was used as the reference standard for tumor stage. Each patient (*n* = 10) provided written informed consent for participation in the study (Table 2). Standard evaluation for these subjects prior to cystectomy includes a CT scan of the abdomen/pelvis with contrast and TURBT. FFPE tissue from the TURBT was obtained. In addition to the standard evaluation, the patients received a non-standard MRI scan of the pelvis, which is considered outside of standard clinical practice and used only for research purposes. Briefly, 4D multiparametric contrast-enhanced MRI [40] was performed on the patients prior to radical cystectomy. The prospective cohort included a balanced distribution of NMIBC (*n* = 5) and MIBC (*n* = 5). NMIBC cases comprised Tis (*n* = 1), Ta (*n* = 1), and T1 tumors (*n* = 3), and MIBC cases comprised T2 (*n* = 5) (Table 4).

### 4.3. Radiologic Staging of BCa

CT and MR images were interpreted independently by two radiologists with special interest in urologic imaging without prior knowledge of the final staging obtained at transurethral resection, cystectomy, or clinical follow-up. If consensus was not reached, a third radiologist reviewed and served as the tie-breaker. Each reviewer assigned a radiologic stage using criteria similar to those previously described in the previous studies [41,42]. Radiologic staging was compared against the final pathological diagnosis from radical cystectomy specimens in all patients.

### 4.4. Radiomics Analysis of CT and MR Images

CT and MR images were provided via Digital Imaging and Communications in Medicine (DICOM) data. Annotations of tumor contours were generated in NIFTI file format using ITK-snap software (version 4.2.0; http://www.itksnap.org (accessed on 10 August 2023)), which defined regions of the tumor to extract radiomic features. PyRadiomics was used to extract imaging features from CT and MR images [43]. The *PyRadiomics* platform can extract radiomic data from medical imaging (such as CT, PET, MRI) using four main steps: (i) loading and preprocessing of the image and segmentation maps; (ii) application of enabled filters; (iii) calculation of features using the different feature classes; and (iv) returning results. Local and global features including intensity variations (histogram-based features), tumor size and shape descriptors, texture features (e.g., gray level co-occurrence matrices, image gradients), and radial features, calculated after applying various combinations of transform-based filters, such as Laplacian of Gaussian (LoG) filters to enhance edge features at different scales and wavelet transforms to capture repetitive patterns at various scales. Features were calculated across voxelized bins. The number of voxelized bins varied between 4, 8, and 16, separately for CT and MR images. For each set of features, quantile normalization was performed, estimating the transformation parameters on the retrospective cohort (MR/CT imaging) and applying the transformation to the prospective cohort (MRI only). In the retrospective cohort, transformed features were correlated with advancing tumor stage as an ordinal predictor, accounting for age and sex as potential confounders, using the limma package’s lmFit function followed by empirical Bayes estimation to reduce the potential for Type I error. MRI/CT radiomic features were selected for further analysis based on a statistical significance cutoff of *p* = 0.05. These candidate features were subsequently evaluated in penalized regression and canonical correlation analyses to reduce the risk of false positives.

### 4.5. RNASeq Data Analysis

RNASeq assay was performed as described in the previous literature [18]. RNASeq gene expression was profiled from TURBT FFPE tissue using the Nextflow nf-core/rnaseq pipeline to generate count matrices from the reads [44]. Sequencing reads were mapped to the hg38 genome—this workflow also included sample quality control (e.g., read trimming, visualization using the MultiQC package, v1.19), alignment and quantification via Salmon, transcript assembly and quantification using StringTie (v3.0.1), and a final quality control check on mapped reads. Read counts were summed across isoforms within gene transcripts and normalized using transcripts per million, DESeq2’s size factor, and the trimmed mean of M-values for downstream analysis, accounting for differences in sequencing depth and RNA composition.

First, a differential expression analysis was conducted using the retrospective cohort for the initial selection of considered sites. The number of genes was reduced to 10,000 based on variance thresholding to limit the number of comparisons. Differential expression analysis based on pseudo-log-transformed expression, accounting for age, sex, and modeling advancing tumor stage as an ordinal predictor, was accomplished using the limma package’s lmFit function followed by empirical Bayes estimation to reduce potential for Type I error. Genes were included for further analysis if their FDR-adjusted *p*-value was less than 0.05. Biological pathways from the Reactome database (v80, released in April 2022) were identified for these statistically significant upregulated and downregulated (based on positive and negative test statistics) using enrichR (v.3.4). From this set, genes were selected for correlative analyses with radiomic features and joint modeling with radiomic features. All RNA-seq assays were performed simultaneously using the same workflow, thereby minimizing the risk of technical batch effects. To account for variability across samples, quantile normalization was applied, and empirical Bayes estimation was used to stabilize variance estimates. Given the limited sample size, additional batch correction strategies (e.g., ComBat) were not required. Statistical significance for differential expression was defined using FDR-adjusted *p*-values (q < 0.05). For visualization in volcano plots, genes surpassing a Bonferroni-adjusted threshold are highlighted.

### 4.6. Identifying Radiomics Features Underlying Prognostic Biologic Pathways of Advanced Stage

After identifying radiomic and gene predictors of advancing tumor stage, we sought to identify radiomic features that underlie gene expression patterns. This was accomplished through sparse canonical correlation analysis (CCA), which is an unsupervised technique for identifying relationships between two sets of high-dimensional variables, in this case, identifying a set of radiomic features that align with a set of genes. For each modality/bin number combination, 4 CCA vectors were identified, corresponding to a panel of genes/radiomic features. For each modality (CT/MRI), CCA components were retained for components that correlated most positively/negatively with advancing stage, determined using Spearman’s correlation. Genes were summarized using enrichR into Reactome pathways as aforementioned, identifying radiomic-based biological pathways tied to tumor progression.

### 4.7. Comparison of Radiogenomic Modeling Approaches for BCa Staging in a Retrospective Cohort for Selection of Radiomics Approach for Signature Development

Next, we sought to compare the ability of CT and MR imaging features to predict the advancing stage in order to decide which modality would be adopted for the prospective cohort analysis. Due to the large number of radiogenomic features, these features were reduced to a simpler composite set of markers using principal component analysis (PCA), with each principal component corresponding to a panel of genes/radiomic features. The top three principal components were correlated with stage using ordinal logistic regression models. These analyses were conducted for all combinations of modality and bin numbers, with and without the addition of the top genes identified in the transcriptome-wide association study. For each combination, the predictiveness of CT and MR features was compared using Akaike and Bayesian information criteria (AIC/BIC), where the lowest value indicates the best fitting model and which modality (CT/MR) and bin number (4, 8, 16) to adopt for prospective analysis. We additionally demonstrated the utility of MR versus CT features through likelihood ratio testing, which separately assessed whether the addition of MR features improved upon the modeling of CT features, and vice versa. Cross-validation was conducted to compare the overall performance of MR radiogenomic features versus CT radiogenomic features within the retrospective cohort. These analyses were conducted across the entire retrospective cohort due to the limited sample size, making this an exploratory analysis for plausible radiogenomic features for further investigation.

### 4.8. Development of Radiogenomics MR Signature in a Retrospective Cohort

After the model comparison between CT and MR features, the best-performing set of radiomic features (MR imaging) was modeled to predict stages in conjunction with genomic features through multivariable analysis. In brief, two ordinal logistic regression models were fit on the retrospective cohort, comprising (1) only the radiomic features and (2) a combined radiogenomics set. Models predicted tumor stage as an ordered outcome and were penalized using Hoseshoe LASSO (least absolute shrinkage and selection operator) in order to account for multicollinearity. Bayesian modeling was used to guard against the potential for Type I error/bias given the small sample size. The importance of radiogenomics features was determined using the LASSO coefficients. Cross-validation within the retrospective cohort was used to validate predictive reliability before prospective testing.

### 4.9. Validation of Radiogenomics Signatures in a Prospective Cohort

The two regression models were validated in a prospective cohort of 10 patients with matching MR and RNASeq data. The models predicted the probability of advanced stage (≥T2) for this group. Receiver operating characteristic (ROC) curves compared the predicted probability P (≥T2) with the outcome. Area under the curve (AUC) estimates were compared between the two models using the median probability estimate from posterior draws, and 95% confidence intervals were calculated using Bayesian bootstrapping, with 4000 posterior AUC samples. In addition to AUC values, 95% confidence intervals were derived using Bayesian bootstrapping. Sensitivity and specificity at the optimal cutoff (Youden’s index) were also reported.

## 5. Conclusions

These early findings illustrating the importance of MR in staging BCa are promising. Additional studies applying more sophisticated machine learning methods, such as the investigation of how genes may further stratify radiomic features by disease progression and vice versa, are underway.

## Figures and Tables

**Figure 1 ijms-26-09570-f001:**
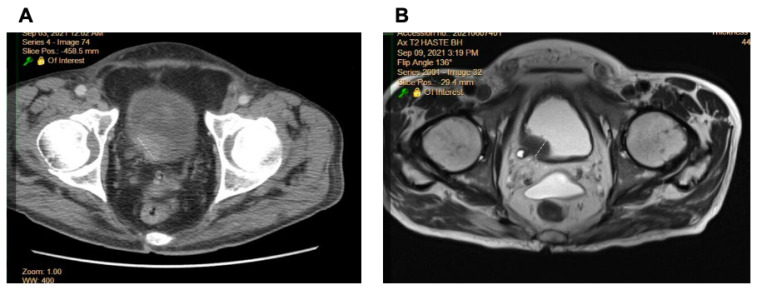
The imaging resolution is lower in CT (**A**) and MRI (**B**). MRI resolution is superior to CT, allowing for the visualization of a tumor in the right posterolateral bladder wall.

**Figure 2 ijms-26-09570-f002:**
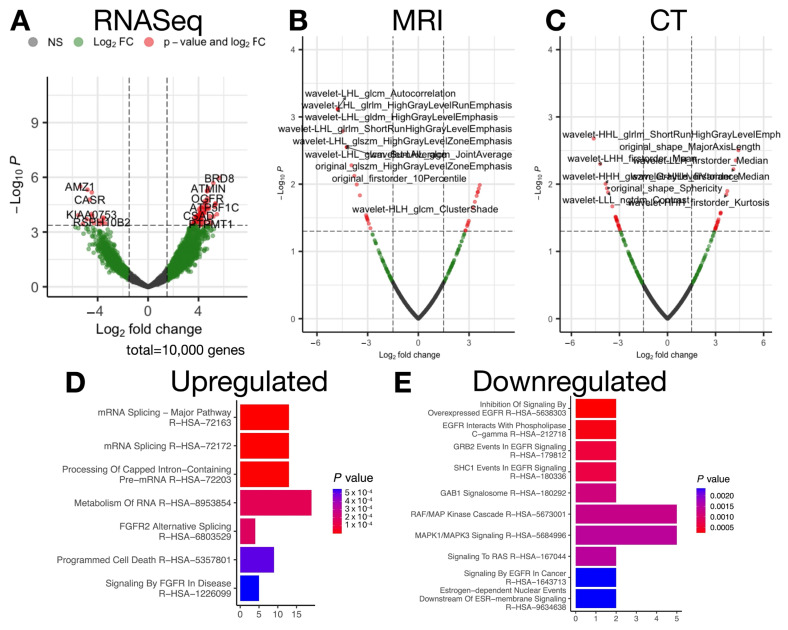
(**A**) Volcano plot of differentially expressed genes from RNASeq, highlighting significantly significant genes (red; beyond Bonferroni cutoff). (**B**,**C**) Volcano plots of radiomic features from MRI (**B**) and CT (**C**), showing significant associations with tumor stage (prior to multiple comparison adjustment). (**D**) Enrichment analysis of upregulated genes reveals pathways related to FGFR2 signaling, RNA splicing, and programmed cell death. (**E**) Enrichment analysis of downregulated genes highlights the suppression of EGFR/RAS/MAPK signaling, suggesting a shift in tumor biology from epithelial proliferation in earlier stages to fibroblast-driven invasion in advanced stages. Together, these plots illustrate that radiomic features and transcriptomic signatures track with known biological programs, FGFR2 and RNA processing in advanced tumors and EGFR/RAS/MAPK signaling in earlier-stage tumors, thereby linking imaging-derived texture changes to plausible molecular pathways.

**Figure 3 ijms-26-09570-f003:**
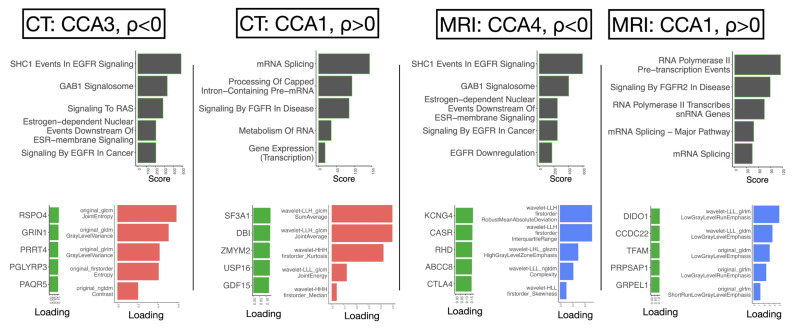
Canonical correlation analysis (CCA) linking radiomic features to gene expression and enriched pathways for CT and MRI. (**Left**) CT-derived CCA3 (ρ < 0) shows a negative correlation with tumor stage, with radiomic features related to entropy and contrast and associated with EGFR signaling pathways. (**Middle-left**) CT-derived CCA1 (positively correlated with tumor stage, ρ > 0) is reflective of FGFR and RNA splicing pathways, with radiomic features including sum average and median intensity. (**Middle-right**) MRI-derived CCA4 (ρ < 0) shows a negative correlation with tumor stage, again associated with EGFR-related pathways, comprising radiomic features linked to skewness and high gray-level emphasis. (**Right**) MRI-derived CCA1 (associated with advanced stage, ρ > 0) is associated with FGFR signaling and RNA processing pathways, with radiomic features related to low gray-level and short-run emphasis, suggesting a shift from EGFR-driven to FGFR-driven tumor biology. CCA thus illustrates how imaging features, such as texture heterogeneity, correspond to biological pathways, supporting the plausibility of FGFR-driven invasion in advanced tumors and EGFR-driven proliferation in earlier-stage disease. Loading values represent the canonical correlation loading coefficients, which indicate the relative contribution of each radiomic or transcriptomic feature to the canonical variates.

**Figure 4 ijms-26-09570-f004:**
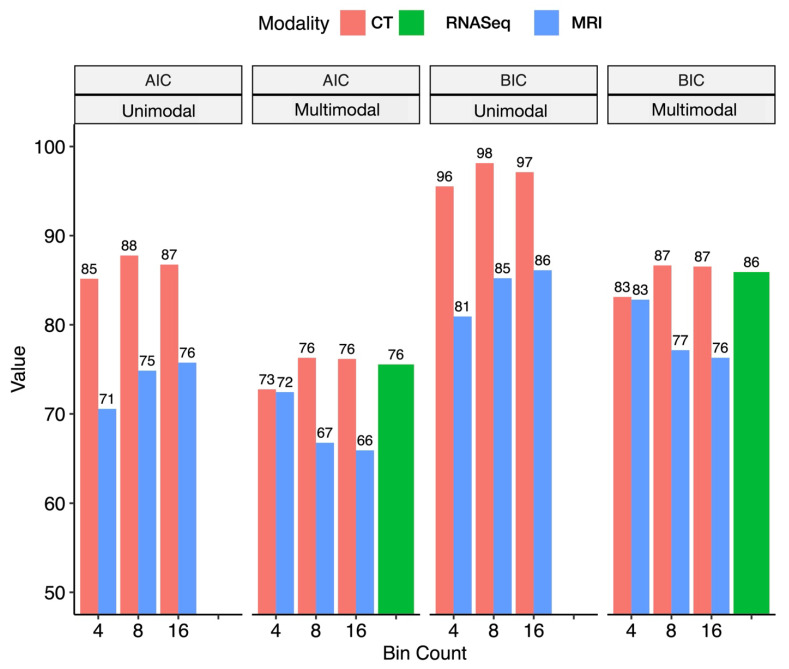
Performance comparison of unimodal (CT or MRI) and multimodal (CT/MRI + RNASeq) models across different bin counts using AIC and BIC criteria. Bars represent values for CT (red), MRI (blue), and RNASeq-only models (green). MRI models consistently show higher performance (lower AIC/BIC) across bin counts, particularly in unimodal settings, while RNASeq integration improves multimodal performance. These results highlight the comparative advantage of MRI over CT for radiogenomic modeling, as MRI features consistently improved model fit, while RNASeq provided incremental value only when combined with MRI, underscoring the robustness of MRI-based approaches for staging.

**Figure 5 ijms-26-09570-f005:**
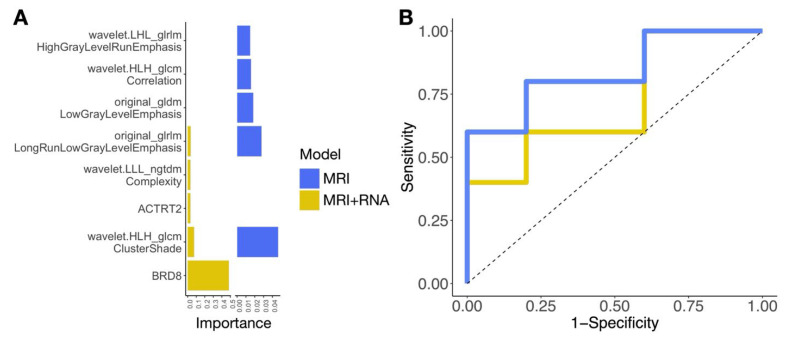
Performance of multivariable models incorporating MRI radiomic features alone (blue) and MRI with RNASeq data (yellow). (**A**) Feature importance plot highlighting the most predictive radiomic and genomic features for each model derived from the retrospective cohort. MRI-based models emphasize radiomic texture features, such as wavelet-HLH_glcm_ClusterShade and wavelet-LHL_glrlm_HighGrayLevelRunEmphasis, while the multimodal model incorporating RNASeq prioritizes gene-level features, such as BRD8 and ACTRT2. (**B**) Receiver operating characteristic (ROC) curves comparing the predictive performance of MRI (blue) and MRI + RNASeq (yellow) models in the prospective cohort, showing differences in sensitivity and specificity for tumor classification.

**Table 1 ijms-26-09570-t001:** (**A**) Performance of a CT scan on the radiologic staging of BCa. (**B**) Performance of MRI on the radiologic staging of BCa.

(**A**)				
	**Pathologic Outcome**			
	**≤pT1**	**pT2**	**≥pT3**	
**Correct**	3	6	5	
**Wrong**	2	6	9	
	**3/5 = 60%**	**6/12 = 50%**	**5/14 = 35%**	**14/31 = 45%**
(**B**)				
	**Pathologic Outcome**			
	**≤pT1**	**pT2**	**≥pT3**	
**Correct**	3	6	8	
**Wrong**	2	6	6	
	**3/5 = 60%**	**6/12 = 50%**	**8/14 = 57%**	**17/31 = 54%**

**Table 2 ijms-26-09570-t002:** Retrospective (four-fold cross-validated, CV) and prospective held-out cohort test set performance statistics; the upper-bound limit for MRI results is 1.0; and sensitivity and specificity were calculated using Youden’s index. Youden’s index was recalculated at every bootstrap iteration for the calculation of standard error (SE) values.

**Retrospective Cohort (Cross-Validation)**			
**Predictors**	**AUC ± SE**	**Sensitivity ± SE**	**Specificity ± SE**
**MRI + RNASeq**	0.80 ± 0.14	0.88 ± 0.13	0.83 ± 0.17
**MRI**	0.77 ± 0.14	0.78 ± 0.2	0.83 ± 0.2
**Prospective Cohort (Held-Out Test)**			
**Predictors**	**AUC ± SE**	**Sensitivity ± SE**	**Specificity ± SE**
**MRI + RNASeq**	0.75 ± 0.22	0.8 ± 0.19	0.78 ± 0.2
**MRI**	0.84 ± 0.22	0.84 ± 0.17	0.89 ± 0.16

**Table 3 ijms-26-09570-t003:** Demographic, clinical, and pathological characteristics of the retrospective study cohort.

Features	BCa (*n* = 31)
Age, years, mean (range)	76.3 (58–88)
<65 years, *n* (%)	4 (12.9)
≥65 years, *n* (%)	27 (87.1)
Male/female ratio (% male)	28:3 (90.3% male)
Race, *n* (%)	
Asian	31 (100)
Primary tumor stage, *n* (%) *	
Intra-vesical (Ta, Tis, or T1–T2)	17 (54.8)
Extra-vesical (T3–T4)	14 (45.2)
Grade, *n* (%)	
Low	0 (0)
High	31 (100)

* Pathologic stage was defined by the final radical cystectomy specimens in all patients.

**Table 4 ijms-26-09570-t004:** Demographic, clinical, and pathologic characteristics of the prospective study cohort.

Features	BCa (*n* = 10)
Age, years, mean (range)	64.1 (21–81)
<65 years, *n* (%)	3 (30)
≥65 years, *n* (%)	7 (70)
Male/female ratio (% male)	6:4 (60% male)
Race, *n* (%)	
White	7 (70)
Hispanic	2 (20)
Asian	1 (10)
Primary tumor stage, *n* (%) *	
NMIBC (Ta, Tis, or T1) **	5 (50)
MIBC (T2) **	5 (50)
Grade, *n* (%)	
Low	1 (10)
High	9 (90)

* Pathologic stage was defined by the final radical cystectomy specimens in all patients. ** NMIBC cases include Tis (*n* = 1), Ta (*n* = 1), and T1 (*n* = 3). MIBC cases include only T2 (*n* = 5).

## Data Availability

Original datasets used in the generation of this manuscript are available from the corresponding author upon reasonable request.

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
