# Peer review of "Comparative Analysis of CT and MRI Combined with RNA Sequencing for Radiogenomic Staging of Bladder Cancer"

_ijms, 2025, doi:10.3390/ijms26199570_

Round 1

Reviewer 1 Report

Comments and Suggestions for Authors

This is an ambitious and well-conceived exploratory study addressing an important and timely clinical question—whether MRI radiomics, combined with RNA sequencing, can improve staging accuracy for muscle-invasive bladder cancer (MIBC) compared to conventional CT-based assessment. The integration of imaging and genomic data in a radiogenomic framework is innovative and aligned with current precision oncology trends. The manuscript is generally well-structured and supported by detailed methodology and clear figures.

However, there are significant limitations in the current study design—particularly small sample size, demographic heterogeneity between cohorts, and potential batch effects—that require more cautious interpretation and stronger discussion. Additionally, some methodological details require clarification, and the presentation of results could be improved for clarity and impact.

Major Comments

  1. Sample Size and Cohort Imbalance

    • The retrospective (n=31, all Asian, higher-stage predominance) and prospective (n=10, mostly Caucasian, earlier-stage predominance) cohorts differ substantially in both demographics and disease stage. These differences may confound the results, particularly in assessing the added value of RNASeq data.

    • Suggest performing sensitivity analyses within more homogeneous subgroups or clearly acknowledging the extent to which cohort differences limit generalizability.

  2. RNASeq Data Quality and Batch Effect

    • The authors note variable FFPE RNA quality and potential batch effects but do not provide quantitative QC metrics or batch correction strategies (e.g., ComBat).

    • Recommend adding details on RNA integrity numbers (RIN), percentage of reads mapped, and whether batch effect correction was attempted.

  3. Statistical Power and Overfitting Risk

    • With such small cohorts, especially in the prospective validation (n=10), complex multivariable and penalized regression models risk overfitting.

    • Suggest reporting model calibration (e.g., calibration plots) and conducting cross-validation within the retrospective set to assess stability of selected features.

  4. Clinical Context and Impact

    • While MRI outperforms CT in staging ≥T3 disease, the added value of radiogenomics in a real-world workflow is less clear, especially given that RNASeq did not improve predictive performance in the prospective cohort.

    • A more explicit discussion of potential clinical implementation barriers (cost, turnaround time, tissue availability) is needed.

  5. Canonical Correlation Analysis (CCA) Interpretation

    • The biological interpretation of CCA components is interesting, but it would help to include explicit examples linking specific radiomic features to histopathological correlates or prior literature.

    • Also, report the statistical significance of these correlations to strengthen confidence.

Minor Comments

  1. Language and Grammar

    • Several minor typographical errors exist (e.g., “Subsequebtly” → “Subsequently”, “patholofical” → “pathological”).

    • Consider careful proofreading for consistency in tense and terminology.

  2. Figures and Tables

    • Figures 1 and 4 appear to partially duplicate content (CT vs MRI resolution comparison). This redundancy could be avoided.

    • Table 1 could benefit from including confidence intervals for staging accuracy.

  3. Abbreviations

    • Ensure all abbreviations are defined at first use in the main text (e.g., “FGFR” and “EGFR” are widely known but should still be introduced for completeness).

  4. Supplementary Data

    • Some important data (Tables S1–S6) are only available as supplementary material; consider moving key findings (e.g., gene lists from differential expression) into the main manuscript.

  5. References

    • The literature review is adequate but could be expanded to include very recent work in deep learning–based MRI bladder staging.

    • I suggest to include in your manuscript this high quality reference related to the role of imaging in Variant histology of MIBC (Bizzarri, F. P., Nelson, A. W., Colquhoun, A. J., & Lobo, N. (2025). Utility of Fluorodeoxyglucose Positron Emission Tomography/Computed Tomography in Detecting Lymph Node Involvement in Comparison to Conventional Imaging in Patients with Bladder Cancer with Variant Histology. European urology oncology, S2588-9311(25)00097-5. Advance online publication. https://doi.org/10.1016/j.euo.2025.03.019)

Reviewer 2 Report

Comments and Suggestions for Authors

The authors compared imaging studies (CT and MRI) and RNA sequencing fingerprints to predict bladder cancer stage.

This is a retrospective study of a small cohort of Japanese patients. The authors aimed to characterize radiogenomic features indicative of advanced tumor stage in bladder cancer. They conclude that MRI was superior to CT scan in predicting advanced tumor stage, which is a reasonable and unsurprising result. They also conclude that radiologic signatures can serve as proxies for key molecular pathways involved in tumor progression. Furthermore, they speculate that radiologic signatures can serve as a non-invasive means of assessing tumor biology.

The manuscript is logical, illustrated with five figures, includes four tables, and cites 31 references.

In the title, the authors indicate that they studied muscle-invasive bladder cancer. The idea of using RNA sequencing fingerprints as a non-invasive substitute to predict tumor stage is novel and attractive. However, some clinical aspects of the studied cohort should be clarified before the manuscript can be accepted for publication.

In Table 3, the authors mention stages Ta, Tis, and T1, which are non-muscle-invasive bladder cancers. It is not clear what method was used to perform clinical staging. The authors mention both TURBT and radical cystectomy in the manuscript. The usual course of treatment includes imaging with a CT scan, with MRI as an option. TURBT is then performed in all eligible patients. The further treatment plan (follow-up or additional therapy) is decided based on the results of the pathological diagnosis after TURBT. It is assumed that the predictive value of CT, MRI, and RNA sequencing data was compared with the pathological diagnosis. If this is the case, the authors should specify in the manuscript which pathological diagnosis (TURBT or cystectomy) was used.

Round 2

Reviewer 1 Report

Comments and Suggestions for Authors

This manuscript addresses an important and timely topic: the use of radiogenomics to improve bladder cancer staging. The integration of CT, MRI, and RNASeq data is innovative, and the authors provide both retrospective and prospective analyses. The writing is generally clear, and the study is well structured. However, several aspects require clarification, methodological strengthening, and improved contextualization before the work can be considered for publication.

Strengths

  1. Novelty: The study combines radiomics with RNASeq to evaluate bladder cancer staging, a relatively unexplored area with high clinical relevance.
  2. Clinical Relevance: Accurate staging of MIBC is crucial for treatment planning; this approach has potential to improve patient stratification.
  3. Comparative Approach: The direct comparison of CT and MRI, both in radiologist assessment and radiogenomic modeling, is valuable.
  4. Pilot Prospective Validation: Including a prospective cohort, albeit small, adds weight to the translational aspect of the work.

Major Concerns

  1. Sample Size and Power:
    • The retrospective and prospective cohorts are small, which limits statistical power. This is acknowledged, but the manuscript could benefit from a more detailed discussion on how sample size impacts the reliability of the findings and what sample size would be needed in future validation studies.
  2. Statistical Analysis:
    • The criteria for feature selection and multiple testing correction (e.g., Bonferroni cutoff) should be more explicitly described.
    • The AIC/BIC comparisons are useful, but model performance metrics (e.g., ROC curves, AUC, sensitivity, specificity with confidence intervals) should be presented in more detail.
  3. Biological Interpretation:
    • While the link between radiomic features and pathways (FGFR, EGFR/RAS/MAPK) is interesting, the discussion could expand on the biological plausibility and relevance of these findings. For example, do these pathways align with known mechanisms of progression in bladder cancer?
  4. Prospective Cohort Analysis:
    • The results suggest that genomic features did not improve the sensitivity and specificity of MRI-based models in the prospective cohort. This is a critical finding that deserves a deeper discussion — why might this be the case? Could it be due to small sample size, heterogeneity, or technical limitations of RNASeq from FFPE?
  5. Clarity in Figures and Tables:
    • Some figures (e.g., Fig. 2–4) are dense and require more explanatory legends.
    • Tables summarizing radiomic and genomic features (currently supplementary) might benefit from a brief synthesis in the main text.

Reference: consider this citations published on EU Oncology related to imaging and BC: doi: 10.1016/j.euo.2025.03.019. 

Minor Comments

  • The English is clear overall but would benefit from light editing to improve flow (e.g., “subsequebtly” → “subsequently”).
  • References to prior radiogenomic studies in bladder cancer should be expanded beyond the four cited papers, if available.
  • The introduction could more explicitly state the hypothesis and how this study advances beyond previous work.

Please ensure consistency in terminology (e.g., "BCa" vs. "bladder cancer").
